# Impact of *BRCA* Mutation Status on Tumor Infiltrating Lymphocytes (TILs), Response to Treatment, and Prognosis in Breast Cancer Patients Treated with Neoadjuvant Chemotherapy

**DOI:** 10.3390/cancers12123681

**Published:** 2020-12-08

**Authors:** Beatriz Grandal, Clémence Evrevin, Enora Laas, Isabelle Jardin, Sonia Rozette, Lucie Laot, Elise Dumas, Florence Coussy, Jean-Yves Pierga, Etienne Brain, Claire Saule, Dominique Stoppa-Lyonnet, Sophie Frank, Claire Sénéchal, Marick Lae, Diane De Croze, Guillaume Bataillon, Julien Guerin, Fabien Reyal, Anne-Sophie Hamy

**Affiliations:** 1Department of Surgery, Institut Curie, University Paris, 75005 Paris, France; beatriz.grandalrejo@curie.fr (B.G.); clemence.evrevin@gmail.com (C.E.); enora.laas@curie.fr (E.L.); isabelle_jardin@hotmail.fr (I.J.); sonia_rozette@hotmail.fr (S.R.); lucielaot@yahoo.fr (L.L.); 2Residual Tumor & Response to Treatment Laboratory, RT2Lab, Translational Research Department, INSERM, U932 Immunity and Cancer, Institut Curie, 26 rue d’Ulm, 75005 Paris, France; elise.dumas@curie.fr (E.D.); anne-sophie.hamy-petit@curie.fr (A.-S.H.); 3Department of Oncology, Institut Curie, 26 rue d’Ulm, 75005 Paris, France; florence.coussy@curie.fr (F.C.); jean-yves.pierga@curie.fr (J.-Y.P.); 4Department of Oncology, Centre René Huguenin, Institut Curie, 35 rue Dailly, 92210 St Cloud, France; etienne.brain@curie.fr; 5Department of Genetics, Institut Curie, 26 rue d’Ulm, 75005 Paris, France; claire.saule@curie.fr (C.S.); Dominique.StoppaLyonnet@curie.fr (D.S.-L.); sophie.frank@curie.fr (S.F.); 6Department of Genetics, Institut Bergonié, 229 Cours de l’Argonne, 33000 Bordeaux, France; c.senechal-davin@bordeaux.unicancer.fr; 7Department of Pathology, Centre René Huguenin, Institut Curie, 35 rue Dailly, 92210 St Cloud, France; marick.lae@curie.fr (M.L.); diane.decroze@curie.fr (D.D.C.); 8Department of Pathology, Centre Henri Becquerel, INSERM U1245, UNIROUEN, University of Normandie, 76038 Rouen, France; 9Department of Pathology, Institut Curie, 26 rue d’Ulm, 75005 Paris, France; guillaume.bataillon@curie.fr; 10Data Office, Institut Curie, 25 rue d’Ulm, 75005 Paris, France; julien.guerin@curie.fr

**Keywords:** BRCA, TILs, pCR, NAC, immunotherapy

## Abstract

**Simple Summary:**

High lymphocytic infiltration (TILs) seem to reflect favorable host antitumor immune responses. In breast cancer, the variation of TILs before and after neoadjuvant chemotherapy (NAC) according to *BRCA* status has been poorly described. Little data is available on their value after treatment. We investigated TIL levels before and after NAC and response to treatment in 267 paired biopsy and surgical specimens. In our study, luminal BCs were associated with pathologic complete response (pCR) and higher TIL levels after chemotherapy completion in patients with *BRCA* pathogenic mutations. Our data supports that (i) NAC should be reconsidered in luminal BCs with *BRCA* pathogenic mutation, (ii) TILs could be a biomarker for response to immune checkpoint blockade in luminal BCs with *BRCA* pathogenic variant who did not achieve a pCR and (iii) exploiting the antitumor immune response in luminal BCs could be an area of active research.

**Abstract:**

Introduction: Five to 10% of breast cancers (BCs) occur in a genetic predisposition context (mainly *BRCA* pathogenic variant). Nevertheless, little is known about immune tumor infiltration, response to neoadjuvant chemotherapy (NAC), pathologic complete response (pCR) and adverse events according to *BRCA* status. Material and Methods: Out of 1199 invasive BC patients treated with NAC between 2002 and 2012, we identified 267 patients tested for a germline *BRCA* pathogenic variant. We evaluated pre-NAC and post-NAC immune infiltration (TILs). Response to chemotherapy was assessed by pCR rates. Association of clinical and pathological factors with TILs, pCR and survival was assessed by univariate and multivariate analyses. Results: Among 1199 BC patients: 46 were *BRCA*-deficient and 221 *BRCA-*proficient or wild type (WT). At NAC completion, pCR was observed in 84/266 (31%) patients and pCR rates were significantly higher in *BRCA-*deficient BC (*p =* 0.001), and this association remained statistically significant only in the luminal BC subtype (*p =* 0.006). The interaction test between BC subtype and *BRCA* status was nearly significant (*P_interaction_* = 0.056). Pre and post-NAC TILs were not significantly different between *BRCA-*deficient and *BRCA-*proficient carriers; however, in the luminal BC group, post-NAC TILs were significantly higher in *BRCA-*deficient BC. Survival analysis were not different between *BRCA-*carriers and non-carriers. Conclusions: *BRCA* mutation status is associated with higher pCR rates and post-NAC TILs in patients with luminal BC. *BRCA*-carriers with luminal BCs may represent a subset of patients deriving higher benefit from NAC. Second line therapies, including immunotherapy after NAC, could be of interest in non-responders to NAC.

## 1. Introduction

Neoadjuvant or pre-operative chemotherapy (NAC) is classically administered to patients with inflammatory or locally advanced breast cancer (BC). Beyond increasing breast-conserving surgery rates [1], it also serves as an in vivo chemosensitivity test and the analysis of residual tumor burden may help understanding treatment resistance mechanisms [2]. In addition, it helps refining the prognosis of patients after NAC, as pathological complete response (pCR) after NAC is associated with a better long-term survival [1,3].

Nearly 5% of breast cancers occur in a context of genetic predisposition, mostly represented by monoallelic pathogenic variants of *BRCA1*, *BRCA2* or *PALB2* genes [4]. Patients with loss-of-function of the *BRCA1* or *2* proteins have a higher cumulated breast cancer risk, with a cumulated lifetime risk at eighty years old of 72% (*BRCA1*) and 69% (*BRCA2*) [5]. The peak incidence for *BRCA1* mutation carriers occurs between 41 and 50 years old (28.3 per 1000 person-years), whereas it occurs ten years later for *BRCA2* mutation carriers (30.6 per 1000 person-years between 51 and 60) [5]. *BRCA1* and *BRCA2* are tumor-suppressor genes that code for proteins involved in homologous recombination (HR) repair. HR deficiency (HRD) occurs when the second allele is inactivated by allelic deletion (often detected by LOH), genic alteration or promoter methylation (for *BRCA1* only). Biallelic *BRCA1/2* inactivation results in genomic instability and theoretically increases the somatic mutational load [6].

Tumors associated with germline or somatic *BRCA1/2* pathogenic mutations display different patterns when compared with sporadic BCs. Cancers occurring among *BRCA1* carriers are more frequently classified as medullary [7], whereas histological subtypes among *BRCA2* carriers tend to be more heterogeneous [8]. In addition, *BRCA1* carriers are more frequently ER-negative, PR-negative and lack *HER2* amplification (i.e., display a triple negative (TNBCs) phenotype [9]) whereas in *BRCA2* carriers, a similar prevalence of ER-positive tumors has been described when compared with sporadic controls [10,11,12,13].

Most of patients with TNBCs receive chemotherapy [14,15]. Due to the alteration of *BRCA1* and *BRCA2* proteins in tumor cells, *BRCA*-mutated cells are unable to properly repair double-strand breaks, classically induced by DNA-alkylating agents [16]. Hence, *BRCA* deficiency has sometimes been associated with a higher sensitivity to platinum agents when compared to other types of neoadjuvant chemotherapy regimens [17,18,19]. In a recent meta-analysis of platinum-based neoadjuvant chemotherapy in TNBC, the addition of carboplatin was not associated with significantly increased pCR rate in *BRCA*-mutated patients (OR = 1.17, CI95% [0.51–2.67], *p* = 0.711) [20]. So far, the benefit of adding a platinum agent in *BRCA*-mutated patients receiving standard neoadjuvant chemotherapy remains a matter of debate. Nevertheless, beyond the controversy upon platinum-based agents in *BRCA*-deficient tumors, the effectiveness of standard NAC in all BC subtypes associated with *BRCA* pathogenic variants compared to controls has been poorly explored so far.

The role of tumor infiltrating lymphocytes (TILs) in BC has been extensively studied over the last decade. High levels of TILs before NAC are associated with higher pCR rates and better survival, especially for TNBC and *HER2*-positive BCs [21,22]. However, despite a growing interest in the field of immunity and oncology, characterization and quantification of TILs across all BC subtypes according to *BRCA* status has not been extensively described. Similarly, no study has evaluated so far, the evolution of immune infiltration after NAC according to *BRCA* status.

The objective of the current study is to determine if pre and post-NAC TILs, chemosensitivity and prognosis differ according to *BRCA* status in a cohort of BC patients treated with NAC.

## 2. Results

### 2.1. Study Population and Tumors Characteristics

The total number of patients included in the neoadjuvant cohort was 1199. Among the whole population, germline *BRCA* pathogenic variant status was available for 267 patients (22.3%), and was not obtained for 932 patients (77.73%, Appendix A). Median age of cohort’s population was 48 years old (range 24–80) and most patients (*n* = 747, 62%) were premenopausal. Median BMI index was 24.74, and 25.8% had direct family history of breast cancer. Patients repartition by subtype was as follows: luminal (*n* = 518, 44%), TNBC (*n* = 376, 31%), *HER2*-positive (*n* = 295, 25%).

Patients with available *BRCA* status were significantly different from patients with *BRCA* status unknown. They were younger, had lower body mass index, were more likely to be diagnosed with grade III, TNBC of no specific type (NST), and to receive standard anthracyclines-taxanes containing regimens than patients not screened (*p* < 0.001) (Table 1, Appendix A).

Among the 267 screened patients, the distribution of *BRCA* status was as follows: *BRCA*-proficient *n* = 221 (83%); *BRCA*-deficient, *n* = 46 (17%) (*BRCA1*-deficient, *n* = 31 (67.39%); *BRCA2*-deficient, *n* = 14 (30.43%) and *BRCA1+2*-deficient, *n* = 1 (2.17%)). Median age at diagnosis for patient with available *BRCA* mutation status was 40 years old (range 24–70) and most patients (*n* = 227, 85%) were premenopausal. Patients repartition by subtype was as follows: luminal (*n* = 90, 33.7%), TNBC (*n* = 110, 41.2%), *HER2*-positive (*n* = 67, 25.1%) (Appendix A, Appendix A).

Carriers of a *BRCA* pathogenic variant were more likely to have familial history of breast cancer (73.9% (34/46) vs. 52.3% (114/218), *p* = 0.012), and to be diagnosed with TNBC (58.7% (27/46) versus 37.6% (83/221), *p* = 0.006) than *BRCA*-proficient patients (Table 1). No other pattern among age, body mass index, histology, tumor size, nor proliferation indices (grade, mitotic index, KI67) was significantly different according to BRCA variant status. These results were substantially similar after the subgroup analysis of BC subtype (Appendix A).

Baseline TILs were available for 192 out of 267 screened patients (72%). Neither pre-NAC str TIL levels (Figure 1A–D), nor IT TILs (Figure 1E–H) were significantly different by *BRCA* status (Appendix A), nor in each BC subtype (Appendix A). There was a strong, positive, linear relationship between stromal and intra-tumoral TILs (Spearman correlation coefficient of 0.74, *p* < 0.001, Appendix A)

### 2.2. Response to Treatment and Post-NAC Immune Infiltration

#### 2.2.1. Response to Treatment

At NAC completion, pCR was observed in 84 out of 266 (31%) patients and pCR rates were significantly different by BC subtype (luminal: 10% (9/89), TNBC: 45% (49/110) and *HER2*-positive 39% (26/67), *p* < 0.001). Pre-NAC str TIL levels were significantly higher in tumors for which pCR was achieved (*p* < 0.001) and there was a significant association between pre-NAC TIL levels and pCR status in the whole population (all: OR = 1.03, CI95% [1.02–1.05], *p* < 0.001; Appendix A) and in the TNBC subgroup (luminal: OR = 1.03, CI95% [1–1.09], *p* = 0.21; TNBC: OR = 1.03; CI95% [1–1.04], *p* = 0.007; *HER2*-positive: OR = 1.02, CI95% [0.99–1.06], *p* = 0.23; Appendix A).

pCR rates were significantly higher in patients with *BRCA*-deficient breast cancers (45.7% (21/46) versus 28% (63/221) in *BRCA*-proficient, *p* < 0.035, Appendix A, Figure 2). After the subgroup analysis of BC subtype, this was confirmed only in the luminal BC subtype (33.3% (5/15), *p* = 0.006), but not in TNBC and *HER2*-positive BCs (48.1% (13/27), *p* = 0.823 and 75% (3/4), *p* = 0.291, respectively, Appendix A, Figure 2). The interaction test between BC subtype and *BRCA* status was nearly significant (Pinteraction = 0.056). There were no differences in pCR rates by *BRCA1* or *BRCA2* mutation status in patients with *BRCA*-deficient tumors (*BRCA1*, 42% (13/31) versus *BRCA2,* 50% (7/14), *p* = 0.7; Appendix A) but the effective of the subpopulations were limited.

However, *BRCA* status was not significantly associated with pCR after multivariate analysis, and only BC subtype (TNBC, OR = 7.14, CI95% [3.39–16.57], *p* < 0.001; *HER2*-positive, OR = 5.64, CI95% [2.5–13.78], *p* = 0.001), tumor size (T2, OR = 0.37, CI95% [0.16–0.83], *p* = 0.017; T3, OR = 0.21, CI95% [0.08–0.55], *p* = 0.002) and pre-NAC str and IT TILs (OR = 1.03, CI95% [1.02–1.05], *p* = 0.001 and OR = 1.04, CI95% [1.02–1.07], *p* = 0.002) were independent predictors of pCR (Appendix A).

#### 2.2.2. Post-NAC Immune Infiltration by BRCA Status

After NAC, str and IT TILs were available in 192 (72%) and 120 (45%) patients respectively. Post-NAC immune infiltration (whether intra-tumoral or stromal) was not significantly different between *BRCA*-deficient and *BRCA*-proficient carriers (Appendix A, Figure 3A–E). However, both str and IT TIL levels were significantly higher in tumors with *BRCA* pathogenic mutations when compared with wild-type tumors in luminal BCs (median str TIL levels: 15% vs. 10%, *p* = 0.009 and median IT TIL levels: 10% vs. 5%, *p* = 0.019, respectively, Appendix A, Figure 3).

Median pre-NAC str TIL were higher than after NAC (20% vs. 10%, 11.95%), also according to *BRCA* status and type (Appendix A, Figure 4). There was no correlation between pre and post NAC str TILs (correlation coefficient of 0.13 and *p* < 0.06, Appendix A) and there was a weak, positive, linear relationship between pre and post NAC IT TIL levels (correlation coefficient of 0.31 and *p* < 0.001, Appendix A).

#### 2.2.3. Survival Analysis

After a median of follow-up of 90.4 months (range from 0.2 to 187 months), 73 patients experienced relapse, and 38 died. RFS and OS were not significantly different between carriers of a *BRCA* pathogenic variant and *BRCA*-proficient patients, neither were they in screened population nor after the subgroup analysis of BC subtype (Appendix A).

## 3. Discussion

In the current study, we did not identify any association between *BRCA* status and immune infiltration, whatever the type of TILs (IT, str). We found a better response to neoadjuvant chemotherapy in tumors associated with a germline *BRCA* pathogenic variant when compared to *BRCA*-WT, however the latter was limited to the restricted group of luminal BCs (*BRCA*-proficient *n* = 75; *BRCA*-deficient, *n* = 15) and was not statistically significant after multivariate analysis, possibly due to the small sample size of the population. Probably in relation, we recovered higher post-NAC lymphocyte infiltration in *BRCA*-deficient tumors in the luminal BC subgroup.

Regarding pre-treatment immune infiltration, Sønderstrup and colleagues [23] analyzed str TIL levels in a nationwide cohort of *BRCA1* and *BRCA2* carriers with primary BCs. They found a greater prevalence of high stromal TILs (defined as TILs-positive tumors with ≥ 60% str TILs) in *BRCA1-*deficient tumors (*n* = 243) when compared with *BRCA2*-deficient tumors (*n* = 168) (36% versus 15% respectively, *p* < 0.0001). However, no control group with *BRCA*-WT tumors was available in this study. In a small study of 85 TNBC patients, Solinas and colleagues [24] investigated the distribution of TILs subpopulations. The tumors of patients in the *BRCA1* or *BRCA2*-mutated group displayed a higher prevalence of TILs-positive tumors (defined as tumors with ≥ 10% str or IT TILs) when compared with the *BRCA-*WT (93.2% versus 75.6% respectively, *p* = 0.037). No other statistically significant differences were identified between *BRCA*-carriers and non-carriers, neither in TILs subpopulations nor their location. More recently, Telli and colleagues [25] investigated the association between TILs, homologous recombination deficiency (HDR) and *BRCA1/2* status in a cohort of 161 TNBC patients pooled from five phase II neoadjuvant clinical trials of platinum-based therapy. They found that IT TILs and str TILs density were not associated with *BRCA1/2* status (*p* = 0.312 and *p* = 0.391, respectively). Consistently with Telli et al., we did not observe any difference in baseline immune infiltration according to *BRCA* status.

Some retrospective studies suggested that tumors displayed higher chemosensitivity according to *BRCA*-mutation status [17,18,19,26,27,28,29,30,31,32]. Arun et al. [30] compared pCR rates after NAC between *BRCA1* or *BRCA2*-carriers (*n* = 57 and *n* = 23, respectively) and WT controls (*n* = 237). The majority of patients (82%) received an anthracycline-taxane containing regimen as NAC. The authors found that *BRCA1* mutation was an independent positive predictor of pCR (OR = 3.16, 95%CI 1.55–6.42, *p* = 0.002). In the largest study so far, Wunderle et al. [18] investigated efficacy of chemotherapy among a cohort of 355 patients composed with 16.6% (59/355) of *BRCA*-carriers. Across all BC subtypes, 64.4% of patients with a *BRCA*1/2 pathogenic variant received anthracycline-based treatments, while the rest received carboplatin. pCR was observed in 54.3% (32/59) of all *BRCA1/2* mutation carriers, and in 39.5% (15/34) of the *BRCA*-carriers versus 13% of the WT BCs in the anthracycline-regimen (Table 2). In our cohort, we found similar results after univariate analysis, and we additionally evidenced a nearly significant interaction with BC subtype. In addition, ongoing trials should determine whether PARP inhibitors might improve outcome when administered in the adjuvant or neoadjuvant setting in early luminal breast cancer patients with *BRCA1/2* mutation [33]. The fact that our results were no longer significant after multivariate analysis is possibly due to a lack of statistical power.

Furthermore, we found that both str and IT TIL levels were higher after NAC completion in the luminal BCs. Whether this difference in post treatment TILs is a cause, a consequence, or unrelated to response to chemotherapy remains unknown. Indeed, post-NAC TIL levels have been shown to be strongly related to response to chemotherapy in BC cohorts including all BC subtypes [34,35,36,37] and response to checkpoint inhibitors (IC) in early TNBC [37]. Moreover, Anurag et al. [38] identified upregulation of the targetable immune-checkpoint components (IDO1, LAG3 and PD1) in AI-resistant luminal B tumors suggesting that luminal BC could also be immunologically “hot”. Besides, only a few studies have investigated the dynamic of TIL levels in response to NAC. Hamy et al. [36] noticed that mean TIL levels decreased after chemotherapy completion across all the BC subtype (pre-NAC TILs: 24.1% vs. post-NAC TILs: 13.0%, *p* < 0.001).

**Table 2 cancers-12-03681-t002:** Literature Review.

Study	Setting/Design	Control Group	Number of Patients (*n*)	TNBC (*n*)	*HER2-*Positive (*n*)	Luminal (*n*)	*BRCA1*	*BRCA2*	*BRCA 1 and 2*	Chemotherapy Regimen	sTILS Evaluation	pCR in BRCA-Carriers vs. Non-Carriers	Survival Analyses	Comments
Byrski (2014) [26] {BCRT	Neoadjuvant epidemiologic prospective cohort	No	10	10	0	10	0	0	0	Cis	No	90%	No	90% (9/10) in *BRCA1*-mutated BC patients achieved a pCR after NAC with cisplatin chemotherapy
Byrski (2015) [27] HCCP	Neoadjuvant epidemiologic prospective cohort	No	107	82	2	NA	107	0	0	Cis	No	61%	No	61% (65/107) in *BRCA1*-mutated BC patients achieved pCR after NAC with cisplatin chemotherapy. In this study of *BRCA1*-mutation carriers, a pCR was also achieved in 56% of 16 patients with ER-positive BC. No survival analysis were provided in the current study.
Hanhnen (2017) [28] JAMA Oncology	Neoadjuvant secondary analysis of the GeparSixto randomized clinical trial	Yes	291	291	0	0	50		0	P + Dox + Bev ± Cb	No	66.7% vs. 36.4%	Yes	Patients with *BRCA*-mutation did not derive a pCR benefit from the addition of carboplatine (65.4% vs. 66.7%) compared to non-*BRCA* carriers (55% vs. 36.4%). No significant difference in overall prognosis observed in the *BRCA*-mutated subgroup.
Sharma (2017) [39] CCR	Neoadjuvant prospective, multicenter, non-randomized trial	Yes	190	190	0	0	30		0	Cb + D	No	59% vs. 56%	No	No significative difference in pCR between *BRCA*-carriers and WT TNBC (59% and 56%, respectively (*p* = 0.83)). The Cb-D regimen was well tolerated and yielded high pCR rates in both *BRCA* associated and WT TNBC. These results are comparable to pCR of previous studies (who investigated pCR after NAC with addition of Cb to AT regimen in TNBC cohort).
Poggio (2018) [20] Annals of Oncology	Neoadjuvant meta-analysis of nine randomized controlled trials	No	96	96	0	0	96		0	P + Dox + Bev ± Cb P + AC ± Cb	No	54.3%	No	Among 96 *BRCA*-mutated patients included in 2 controlled trials, the addition of carboplatin was not associated with increased pCR rate (OR 1.17, 95% CI 0.51–2.67, *p* = 0.711). No survival analyses were available according to *BRCA* status.
Telli (2019) [25] CCR	Five randomized controlled trials	Yes	161	161	0	0	34		0	Cb + Gem + Iniparib; Cis; Cis + Bev; Cb + Eribulin; Cb + nab-P ± Vorinostat	Yes	No	No	pCR was achieved in 51 (31.7%) patients. In patients with TNBC treated with neoadjuvant platinum-based therapy, iTIL and sTIL densities were not significantly associated with *BRCA1/2*-mutated tumor status (*p* = 0.312 and *p* = 0.391). In multivariate analyses, sTIL density (OR 1.23, 95% CI 0.94–1.61, *p* = 0.139) was not associated with pCR, but was associated with RCB 0/I status (OR 1.62, 95% CI 1.20–2.28, *p* = 0.001).
Sønderstrup (2019) [23] Acta Oncologica	Epidemiologic prospective mulitcentric cohort (nationwide)	No	411	NA	24	NA	243	168	0	NA	Yes	No	Yes	High sTILs (defined as TILs > 60%) were observed in 36% in *BRCA1*- and 15% in *BRCA2-*mutated tumors (*p* < 0.0001). Significant association with survival (OS and DFS) was observed in *BRCA1* subgroup. sTILs are an important prognostic factor in *BRCA* BC and increasing sTILs is associated with a better prognosis.
Byrski (2009) [17] JCO	Neoadjuvant Epidemiologic epidemiologic retrospective cohort	No	102	NA	6	NA	102	0	0	CMF; AT; AC FAC or Cis	No	23.5%	No	pCR was achieved in 23.5% of 102 patients with a *BRCA1* mutation who received NAC. Especially, a complete pCR was observed in 8% (2/25) with AT- regimen (standard of care) compared to 83% (10/12) with cisplatin.
Chappuis (2002) [29] JMG	Neoadjuvant Retrospective retrospective multicentric clinical trial	Yes	38	NA	NA	NA	7	4	0	FAC; AC; CEF AC + CMF AC + D	No	44% vs. 4%	No	pCR was achieved in 44% (4/11) of the *BRCA*-carriers and 4%(1/27) of the non-carriers (*p* = 0.009). No survival analysis were experienced in this study.
Arun (2011) [30] JCO	Neoadjuvant Epidemiologic epidemiologic retrospective cohort	Yes	317	77	60	NA	57	23	0	A-single agent; AT or T-single-agent	No	46% vs. 22%	Yes	pCR was achieved in 46% of *BRCA1*-carriers and 13% of *BRCA2*-carriers and 22% of *BRC*A non-carriers (<0.001). In the multivariate logistic model, *BRCA*1 status (OR = 1.96, *p* = 0.03) remained as independant significant predictors of a pCR. No significant difference in overall prognosis.
Wang (2014) [40] Annals of Oncology	Neoadjuvant Epidemiologic retrospective cohort	Yes	652	652	0	0	52	NA	0	A-single agent; AT or T-single-agent	No	53.8% vs. 29.7%	Yes	The pCR rate was 31.6% in the 652 patients who received NAC. *BRCA1* carriers had a significantly higher pCR rate than non-carriers (*BRCA1* carriers versus non-carriers, 53.8% versus 29.7%, *p* < 0.001). Among women treated with anthracycline with or without taxane regimens, the pCR rate was 57.1% for *BRCA1* carriers, 29.0% for non-carriers (*p* < 0.001). The RFS was similar according to *BRCA* status.
Paluch-Shimon(2016) [31] BCRT	Neoadjuvant epidemiologic retrospective cohort	Yes	80	80	0	0	34	0	0	AT	No	68% vs. 37%	Yes	The *BRCA1*-carriers had pCR rate of 68% compared with 37% among non-carriers, *p* = 0.01. Yet this did not translate into superior survival for *BRCA1* carriers compared with non-carriers.
Bignon (2017) [41] Breast	Neoadjuvant epidemiologic retrospective cohort	No	53	53	0	0	46	6	1	A-single agent or AT	No	66%	Yes	The pCR rate was 38.3% [95% CI, 26%–55%] among *BRCA1* mutation carriers, and 66% among the 6 *BRCA2* mutation carriers. 15 relapses and 6 s cancers were recorded during the follow-up period. 11 deaths occurred, all of which were in the non-pCR group. DFS (*p* < 0.01) and OS (*p* < 0.01) were significantly better in the pCR group than the non-pCR group.
Wunderle (2018) [18] BCRT	Neoadjuvant Epidemiologic retrospective cohort	Yes	355	138	58	159	43	16	0	AT; Cb	No	54.3% vs. 12.6%	Yes	pCR was observed in 54.3% of *BRCA1/2* mutation carriers, but only in 12.6% of non-carriers. The adjusted odds ratio was 2.48 (95% CI 1.26–4.91) for *BRCA1/2* carriers versus non-carriers. No difference in overall survival was observed.
Saether (2018) [32] HCCP	Neoadjuvant Epidemiologic retrospective cohort	No	12	NA	NA	NA	12	0	0	Cis + Dox or Cb + D	No	83%	No	11 patients received a combination of cisplatin and doxorubicin, and 1 patient received carboplatin and docetaxel. 83% (10/12) of the *BRCA1*-carriers achieved pCR. This results were comparable to existing results found in similar studies. No information about BC subtype among the study population and the toxicity of the chemotherapy was not evaluated.
Sella (2018) [19] Breast	Neoadjuvant Epidemiologic retrospective cohort	Yes	43	43	0	0	14	0	0	AT ± Cb	No	67% vs. 38%	No	pCR was achieved in 38% in *BRCA* WT compared to 67% in *BRCA*-associated TNBC (*p* = 0.232). No benefit from the addition of carboplatine in *BRCA*-carriers (64.3% vs. 67%) compared to non-*BRCA* carriers (44.8% vs. 38%) when compared to historic institutional rates with AT.
Solinas (2019) [24] Cancer Letters	Epidemiologic retrospective cohort	Yes	85	85	0	0	38	6	0	NA	Yes	No	Yes	The *BRCA*-mutated tumors had a significantly higher incidence of TIL-positive levels compared to WT (44% and 41%, respectively *p* = 0.037). No significant difference between *BRCA*-mutated and WT groups neither in TIL subpopulation nor their location. No difference in I-DFS and OS after stratification on TIL infiltration levels.
Our study (2020)	Epidemiologic retrospective cohort	Yes	267	110	67	90	31	14	1	A-single agent; AT or T-single-agent	Yes	45.7% vs 28%	Yes	Among the whole population, 84 tumors achieved a pCR (31.5%). After stratification by BC subtype, pCR rates were significantly higher in luminal *BRCA*-mutated BCs when compared with WT tumors (33.3% vs. 5.4%, *p* = 0.006).Pre and post-NAC str or IT TILs were not significantly different between *BRCA*-carriers and non-carriers in whole population. In the luminal BC, both str and IT post-NAC TIL levels were significantly higher in *BRCA*-mutated tumors when compared with WT tumors but was no longer significant after multivariate analysis. No difference in RFS or OS between *BRCA*-mutated and *BRCA*-WT patients.

Abbreviations: CMF = cisplatine-methotrexate-fluorouracile; AT = doxorubicine-docetaxel; AC = doxorubicine-cyclophosphamide; FAC = fluorouracile-doxorubicin-cyclophophosphamide; CEF = cyclophosphamide-epirubicine-fluorouracile; A = anthracycline; Dox = doxorubicine; D = docetaxel; Cb = carboplatin; Cis = cisplatine; Bev = bevacizumab; Gem = gemcitabine; Nab-P = nabpaclitaxel; P = paclitaxel; T = taxane.

This decrease was strongly associated with high pCR rates, and the variation of TIL levels was strongly inversely correlated with pre-NAC TIL levels (and the variation of TIL levels was strongly inversely correlated with pre-NAC TIL levels (r = −0.80, *p* < 0.001).

Finally, in line with several recently published clinical studies [42,43,44], we found that survival outcomes were not different between *BRCA*-carriers and non-carriers. A multivariate study, including 223 BC patients carrying *BRCA* pathogenic variants and 446 controls with sporadic BC matched for age and year of diagnosis, showed no difference in terms of specific BC survival between *BRCA1* or *BRCA2* mutation carriers and controls [45]. Templeton et al. evaluated a total of 16 studies comprising data from 10,180 patients and concluded that *BRCA* pathogenic mutations were not associated with a worse overall survival [46]. The difference between the pCR rate and survival analysis could be due to several factors. First, *BRCA* mutation carriers are commonly offered additional treatment, including a bilateral mastectomy. Second, the increase of TILs in the surgical piece might reveals a higher immunogenic tumor, which may involve a more sustained response to treatment over time. Besides, carriers of a *BRCA* pathogenic variant were more likely to be diagnosed with TNBC. Copson et al. [44] have shown that *BRCA* mutation carriers with triple-negative breast cancer might have a survival advantage during the first few years after diagnosis compared with non-carriers. This benefit might reflect greater sensitivity of *BRCA*-mutant breast cancers to chemotherapy or the greater visibility to host immune attack.

Limits of our study include its retrospective observational design as well as small effectives potentially leading to a lack of statistical power. Therefore, our results might be submitted to evaluation biases, especially for the time-to-event analysis. Indeed, we present a study of patients with two rare conditions. First, according to French national guidelines, neoadjuvant chemotherapy is currently prescribed only in 15% of the patients with locally advanced breast cancers. Second, screening of inherited *BRCA* mutation is performed in a highly selected population representing nearly one quarter of breast cancer [47]. Our study design does not allow us to draw firm conclusions and future studies are warranted to confirm the hypotheses generated. Moreover, the incidence of bi-allelic pathogenic alterations in HR-related genes according to somatic origin is well-known and ranches from 1 to 2% [48] but we did not explore somatic mutational status in the tumor tissues in the current study. The study also has several strengths, for instance from being the largest cohort with a *BRCA*-WT control group, and analyses performed after stratification by BC subtype. Finally, to our knowledge, we provide data on post-NAC immune infiltration according to *BRCA* status for the first time.

## 4. Materials and Methods

### 4.1. Patients and Tumors

The study was performed on a retrospective institutional cohort of 1199 female patients with T1-T3NxM0 invasive BC (NEOREP Cohort, CNIL declaration number 1547270) treated with NAC at Institute Curie (Paris and Saint-Cloud) between 2002 and 2012. The cohort included unifocal, unilateral, non-recurrent, non-metastatic tumors, excluding T4 tumors (inflammatory, chest wall or skin invasion). Approved by the Breast Cancer Study Group of Institute Curie, the study was conducted according to institutional and ethical rules concerning research on tissue specimens and patients. Informed consent from patients was not required.

Information on family history, clinical characteristics (age; menopausal status; body mass index) and tumor characteristics (clinical tumor stage and grade; histology; clinical nodal status; ER, PR and HER2 status; BC subtype; mitotic index; Ki67) were retrieved from electronic medical records. All the patients received NAC, and additional treatments were decided according to national guidelines (see Appendix A).

### 4.2. Tumors Samples

In accordance with French national guidelines [49], cases were considered estrogen receptor (ER)-positive or progesterone receptor (PR)-positive if at least 10% of tumor cells expressed estrogen and/or progesterone receptors (ER/PR), and endocrine therapy was prescribed when this threshold was exceeded. HER2 negative status was defined as 0 or 1 + on immunohistochemistry (IHC) stained tissue section. IHC 2+ scores were subsequently analyzed by fluorescence in situ hybridization (FISH) to confirm *HER2* positivity. Pathological BC were classified into subtypes (TNBC, *HER2*-positive, and luminal *HER2*-negative [referred to hereafter as “luminal”]) (see Appendix A).

### 4.3. TIL Levels, Pathological Complete Response and Pathological Review

TIL levels were evaluated retrospectively for research purposes, by two pathologists (ML and DdC) specialized in breast cancer. TIL levels were assessed on formalin-fixed paraffin-embedded (FFPE) tumor tissue samples from pretreatment core needle biopsies and the corresponding post-NAC surgical specimens, according to the recommendations of the international TILs Working Group before [50] and after NAC [51]. TILs were defined as the presence of a mononuclear cell infiltrate (including lymphocytes and plasma cells, excluding polymorphonuclear leukocytes). TILs in direct contact with tumor cells were counted as intra-tumoral TILs (IT TILs) and those in the peri-tumoral areas as stromal TILs (str TILs). They were evaluated both in the stroma and within tumor scar border, after excluding areas around ductal carcinoma in situ, tumor zones with necrosis and artifacts, and were scored continuously as the average percentage of stroma area occupied by mononuclear cells. We defined pathological complete response (pCR) as the absence of invasive residual tumor from both the breast and axillary nodes (ypT0/is N0).

### 4.4. BRCA Status

Genetic counseling was offered based on individual or family criteria (see Appendix A). When constitutional genetic analysis of *BRCA1* and *BRCA2* genes were required, Denaturing High Performance Liquid Chromatography (DHPLC) and Sanger sequencing were performed to search for point alterations, and Quantitative Multiplex Polymerase Chain Reaction of Short Fluorescent (QMPSF) to research large gene rearrangements between 2002 and 2012. In case of previously known pathogenic familial variants, targeted tests were performed.

### 4.5. Survival Endpoints

Relapse-free survival (RFS) was defined as the time from surgery to death, loco-regional recurrence or distant recurrence, whichever occurred first. Overall survival (OS) was defined as the time from surgery to death. For patients for whom none of these events were recorded, data was censored at the time of last known contact. Survival cutoff date analysis was 1 February 2019.

### 4.6. Statistical Analysis

Pre- and post-NAC TIL levels were analyzed as continuous variables. All analyses were performed on the whole population and after stratification by BC subtype. To compare continuous variables among different groups, Wilcoxon-Mann-Whitney test was used for groups including less than 30 patients and for variables displaying multimodal distributions; otherwise, student t-test was used. Association between categorical variables was assessed with chi-square test, or with the Fisher’s exact test if at least one category included less than three patients. In boxplots, lower and upper bars represented the first and third quartile respectively, the medium bar was the median, and whiskers extended to 1.5 times the inter-quartile range. Factors predictive of pCR were introduced in a univariate logistic regression model. Covariates selected for multivariate analysis were those with a *p*-value no greater than 0.1 after univariate analysis. Survival probabilities were estimated by Kaplan-Meyer method, and survival curves were compared with log-rank tests. Hazard ratios (HR) and their 95% confidence intervals (CI) were calculated with the Cox proportional hazard model. Analyses were performed with R software version 3.1.2(RStudio Team (2018). RStudio Integrated Development for R.RStudio, Inc., Boston, MA URL)). The significance threshold was set at 5%.

## 5. Conclusions

Although the number of patients *BRCA*-deficient is restrained, our study raises several hypotheses. First, it generates an unprecedented hypothesis that luminal BC patients with germline *BRCA* pathogenic variants might represent a subset of luminal BCs that are more likely to benefit from chemotherapy as primary treatment than the whole luminal BC population. It is known that the absolute benefit of chemotherapy is lower in luminal BC than in the other BC subtypes [52]. Genetic signatures have been implemented in the daily clinical practice to complement classic prognostic factors and aid in treatment decisions with luminal-*HER2* negative early-stage breast cancer [53]. While providing information on the expression of some genes related to estrogen receptor, proliferation and immunity, such tools enable a better prediction of the response to standard neoadjuvant chemotherapy. Nowadays, genomic tests allow us to scale down or escalate treatments in luminal-*HER2* negative early breast cancer with intermediate prognostic factors. If further validated in independent cohorts, germline breast cancer *BRCA* mutation could be in the future, in a luminal context, an argument to boost a patient to chemotherapy, in addition to multigene assays. Second, patients not achieving pCR may be candidates for post-operative clinical trials exploring alternative therapeutic strategies. As post-NAC immune infiltration seems to be higher in post-NAC specimens of luminal tumors with *BRCA* pathogenic mutations, we can hypothesize that those tumors would be more likely to respond to checkpoint inhibitors after chemotherapy. Second line trials using immune checkpoint inhibitors (such as anti–PD-1 and anti–PD-L1 antibodies) alone or in combination, together with endocrine therapy could be a relevant strategy for patients failing to reach pCR at NAC completion.

## Figures and Tables

**Figure 1 cancers-12-03681-f001:**
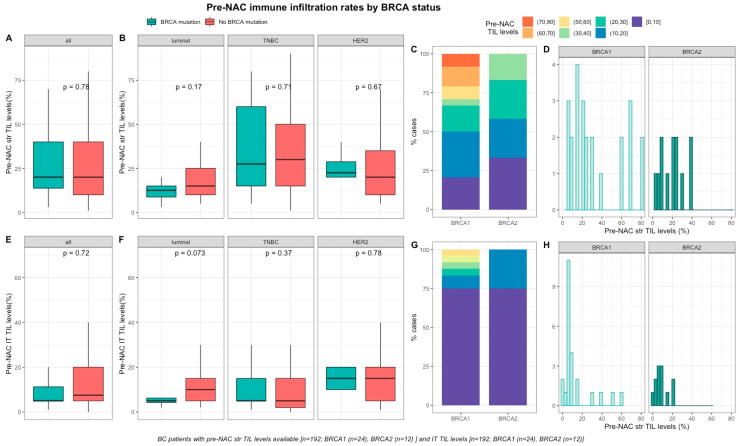
Associations between pre-NAC TILs and *BRCA* status in whole population, and by breast cancer subtype. Bottom and top bars of the boxplots represent the first and third quartiles, respectively, the medium bar is the median, and whiskers extend to 1.5 times the interquartile range. (**A**) stromal lymphocytes among the whole population (All (*n* = 192), *BRCA* mutation (*n* = 36), *BRCA* wild-type (*n* = 156)). (**B**) stromal lymphocytes in each BC subtype (Luminal (*n* = 52), *BRCA* mutation (*n* = 8), *BRCA* wild-type (*n* = 44); TNBC (*n* = 97), *BRCA* mutation (*n* = 24), *BRCA* wild-type (*n* = 73); *HER2* (*n* = 43), *BRCA* mutation (*n* = 4), *BRCA* wild-type (*n* = 39)). (**C**) percentage of tumor according to pre-NAC stromal lymphocytes levels binned by 10% increment in patients with *BRCA*-deficient (*BRCA*1 (*n* = 24), *BRCA*2 (*n* = 12)). (**D**) distribution of pre-NAC stromal lymphocytes by gene mutations (histogram plot) in patients with *BRCA*-deficient (*BRCA*1 (*n* = 24), *BRCA*2 (*n* = 12)). (**E**) intratumoral lymphocytes among the whole population (All (*n* = 192), *BRCA* mutation (*n* = 36), *BRCA* wild-type (*n* = 156)). (**F**) intratumoral lymphocytes in each BC subtype (Luminal (*n* = 52), *BRCA* mutation (*n* = 8), *BRCA* wild-type (*n* = 44); TNBC (*n* = 97), *BRCA* mutation (*n* = 24), *BRCA* wild-type (*n* = 73); *HER2* (*n* = 43), *BRCA* mutation (*n* = 4), *BRCA* wild-type (*n* = 39)). (**G**) Percentage of tumor according to pre-NAC intratumoral lymphocytes levels binned by 10% increment in patients with *BRCA*-deficient (*BRCA*1 (*n* = 24), *BRCA*2 (*n* = 12)). (**H**) distribution of pre-NAC intratumoral lymphocytes by gene mutations (histogram plot) in patients with *BRCA*-deficient (*BRCA*1 (*n* = 24), *BRCA*2 (*n* = 12)).

**Figure 2 cancers-12-03681-f002:**
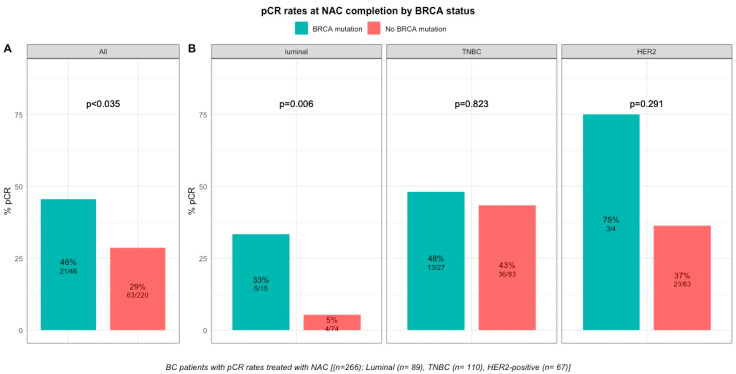
Barplot of associations between response to treatment and *BRCA* status in whole population, and by breast cancer subtype. (**A**), among the whole population (All (*n* = 266), *BRCA* mutation (*n* = 46), *BRCA* wild-type (*n* = 220)). (**B**), by BC subtype (Luminal (*n* = 89), *BRCA* mutation (*n* = 15), *BRCA* wild-type (*n* = 74); TNBC (*n* = 110), *BRCA* mutation (*n* = 27), *BRCA* wild-type (*n* = 83); HER2 (*n* = 67), *BRCA* mutation (*n* = 4), *BRCA* wild-type (*n* = 63)).

**Figure 3 cancers-12-03681-f003:**
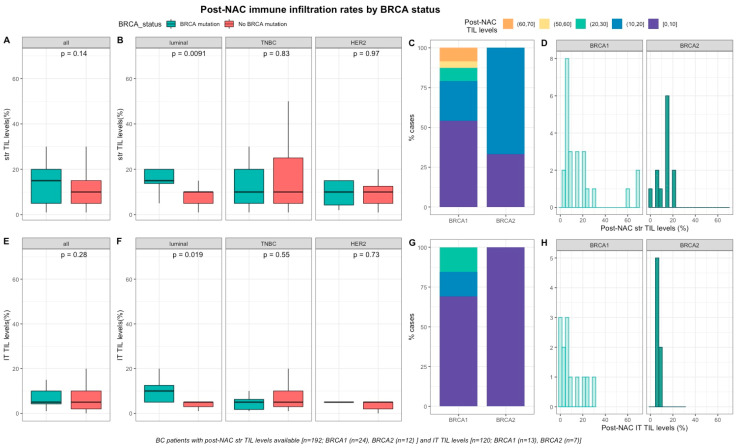
Associations between post-NAC TILs and *BRCA* status in whole population, and after stratification by breast cancer subtype. Bottom and top bars of the boxplots represent the first and Table 1. 5 times the interquartile range. (**A**) stromal lymphocytes among the whole population (All (*n* = 192), *BRCA* mutation (*n* = 36), *BRCA* wild-type (*n* = 156)). (**B**) stromal lymphocytes in each BC subtype (Luminal (*n* = 52), *BRCA* mutation (*n* = 8), *BRCA* wild-type (*n* = 44); TNBC (*n* = 97), *BRCA* mutation (*n* = 24), *BRCA* wild-type (*n* = 73); *HER2* (*n* = 43), *BRCA* mutation (*n* = 4), *BRCA* wild-type (*n* = 39)). (**C**) Percentage of tumor according to post-NAC stromal lymphocytes levels binned by 10% increment in patients with *BRCA*-deficient (*BRCA*1 (*n* = 24), *BRCA*2 (*n* = 12)). (**D**) distribution of post-NAC stromal lymphocytes by gene mutations (histogram plot) in patients with *BRCA*-deficient (*BRCA*1 (*n* = 24), *BRCA*2 (*n* = 12)). (**E**) intratumoral lymphocytes among the whole population (All (*n* = 120), *BRCA* mutation (*n* = 20), *BRCA* wild type (*n* = 100)). (**F**) intratumoral lymphocytes in each BC subtype (Luminal (*n* = 44), *BRCA* mutation (*n* = 7), *BRCA* wild-type (*n* = 37); TNBC (*n* = 50), *BRCA* mutation (*n* = 12), *BRCA* wild-type (*n* = 38); *HER2* (*n* = 26), *BRCA* mutation (*n* = 1), *BRCA* wild-type (*n* = 25)). (**G**) percentage of tumor according to post-NAC intratumoral lymphocytes levels binned by 10% increment in patients with *BRCA*-deficient (*BRCA*1 (*n* = 13), *BRCA*2 (*n* = 7)). (**H**) distribution of post-NAC intratumoral lymphocytes by gene mutations (histogram plot) in patients with *BRCA*-deficient (*BRCA*1 (*n* = 13), *BRCA*2 (*n* = 7)).

**Figure 4 cancers-12-03681-f004:**
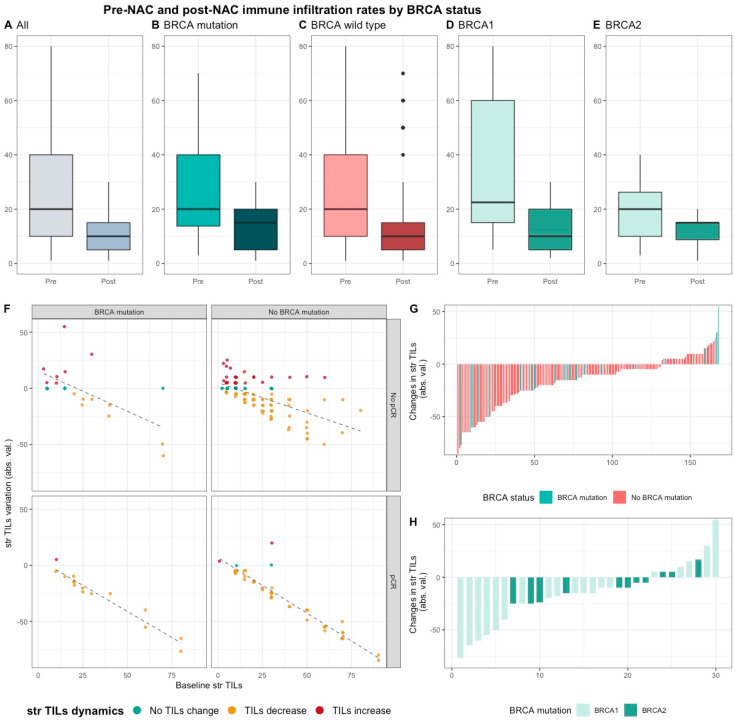
Pre-NAC and post-NAC stromal immune infiltration rates in the whole population and by *BRCA* status. (**A**–**E**) bar plots of str TIL levels before and after NAC in the whole population and in *BRCA* pathogenic variant. Bottom and top bars of the boxplots represent the first and third quartiles, respectively, the medium bar is the median, and whiskers extend to 1.5 times the interquartile range. (All (*n* = 192); *BRCA* mutation (*n* = 36), *BRCA* wild-type (*n* = 156); *BRCA*1 (*n* = 24), *BRCA*2 (12)). (**F**) variation of str TIL levels according to the pre-NAC str TIL levels binned by *BRCA* status and response to chemotherapy. Points represent the difference between pre- and post-NAC paired TIL levels values of a given patient and are colored according to TIL variation category (TIL level decrease: yellow/no change: green/increase: red) (All (*n* = 191), *BRCA* mutation (*n* = 36), *BRCA* wild-type (*n* = 155)). (**G**–**H**) waterfall plot representing the variation of TIL levels according to *BRCA*-deficient (*BRCA*1-deficient, *BRCA*2-deficient); each bar represents one sample, and samples are ranked by increasing order of TIL level change. Paired samples for which no change was observed have been removed from the graph. (All (*n* = 191), *BRCA* mutation [(*n* = 36), *BRCA*1, *n* = 24; *BRCA*2 = 12)], *BRCA* wild-type (*n* = 155)).

**Table 1 cancers-12-03681-t001:** Patients’ characteristics among the whole population.

Characteristics	Class	All	*BRCA* Mutation	*BRCA* Wild-Type	Not Screened	*p*
		*n* = 1199 (100%)	*n* = 46 (3.8%)	*n* = 221 (18.4%)	*n* = 932 (77.7%)	
Age (mean)		48.6	39.5	41.7	50.6	<0.01
Menopausal Status	pre	747 (62.8)	41 (89.1%)	187 (85.0%)	519 (56.2%)	<0.01
	post	442 (37.2)	5 (10.9%)	33 (15.0%)	404 (43.8%)	
BMI (mean)		24.7	22.8	23.6	25.1	<0.01
BMI class	(15,19]	72 (6.0)	6 (13.3)	17 (7.7)	49 (5.3)	<0.01
	(19,25]	664 (55.7)	31 (68.9)	147 (66.5)	486 (52.4)	
	(25,30]	299 (25.1)	4 (8.9)	43 (19.5)	252 (27.2)	
	(30,50]	158 (13.2)	4 (8.9)	14 (6.3)	140 (15.1)	
Family history of BC	no	887 (74.2)	12 (26.1%)	104 (47.7%)	771 (82.7%)	<0.01
	yes	309 (25.8)	34 (73.9%)	114 (52.3%)	161 (17.3%)	
Clinical tumor size	T1	70 (5.8%)	5 (10.9%)	22 (10.0%)	43 (4.6%)	<0.01
	T2	798 (66.6%)	28 (60.9%)	153 (69.2%)	617 (66.3%)	
	T3	330 (27.5%)	13 (28.3%)	46 (20.8%)	271 (29.1%)	
Clinical	N0	525 (43.8%)	17 (37.0%)	93 (42.1%)	415 (44.6%)	0.51
nodal status	N1-N2-N3	673 (56.2%)	29 (63.1%)	128 (57.9%)	516 (55.4%)	
Histology	NST	1062 (90%)	43 (93.5%)	213 (96.4%)	806 (88.3%)	0.03
	others	118 (10%)	3 (6.5%)	8 (3.6%)	108 (11.6%)	
Grade	I-II	479 (41.4%)	10 (23.3%)	76 (34.7%)	393 (43.9%)	0.01
	III	678 (58.6%)	33 (76.7%)	143 (65.3%)	502 (56.1%)	
Mitotic Index (mean)		25.1	30.8	25.6	24.6	0.25
Subtype	luminal	528 (44.0%)	15 (32.6%)	75 (33.9%)	438 (47.0%)	<0.01
	TNBC	376 (31.4%)	27 (58.7%)	83 (37.6%)	266 (28.5%)	
	*HER2*	295 (24.6%)	4 (8.7%)	63 (28.5%)	228 (24.5%)	
str TILs (mean)		20.0 [10.0–30.0]	20.0 [13.8–40.0]	20.0 [10.0–40.0]	15.0 [10.0–30.0]	0.02
IT TILs (mean)		5.0 [5.0–15.0]	5.0 [5.0–11.2]	7.5 [5.0–20.0]	5.0 [3.0–15.0]	0,47
NAC Regimen	AC	235 (19.6%)	4 (8.7%)	25 (11.4%)	206 (22.2%)	<0.01
	AC-Taxanes	845 (70.7%)	41 (89.1%)	180 (81.8%)	624 (67.1%)	
	Taxanes	25 (2.1%)	1 (2.2%)	6 (2.7%)	18 (1.9%)	
	Others	91 (7.6%)	0 (0.0%)	9 (4.1%)	82 (8.8%)	
pCR class	No pCR	911 (76.2)	25 (54.3)	157 (71.4)	729 (78.4)	<0.001
	pCR	285 (23.8)	21 (45.7)	63 (28.6)	201 (21.6)	
Nodal involvment	0	682 (57.0)	35 (76.1)	141 (64.1)	506 (54.4)	0.003
	1–3	341 (28.5)	6 (13.0)	58 (26.4)	277 (29.8)	
	≥4	174 (14.5)	5 (10.9)	21 (9.5)	148 (15.9)	
str TILs (mean)		10.0 [5.0–15.0]	15.0 [5.0–20.0]	10.0 [5.0–15.0]	10.0 [5.0–15.0]	0.36
IT TILs (mean)		5.0 [2.0–10.0]	5.0 [4.3–10.0]	5.0 [2.0–10.0]	5.0 [2.0–10.0]	0.57

Missing data: Menopausal status, *n* = 10; BMI (continuous), *n* = 6; BMI class, *n* = 6; Family history, *n* = 3; Clinical tumor stage, *n* = 1; Clinical nodal status, *n* = 1; Histology, *n* = 19; Grade, *n* = 42; Mitotic index, *n* = 502; Pre-NAC str TILs, *n* = 482; Pre-NAC IT TILs, *n* = 482; NAC regimen, *n* = 3; pCR status, *n* = 3; Post-NAC Nodal involvment, *n* = 2; Post-NAC str TILs, *n* = 482; Post-NAC IT TILS, *n* = 714. *Abbreviations:* NAC = neoadjuvant chemotherapy; BMI = body mass index; NST = no special type; TNBC = triple negative breast cancer; str TILs = stromal tumor-infiltrating lymphocytes; IT TILs = intratumoral-infiltrating lymphocytes; AC = anthracyclines; pCR = Pathologic complete response. The “*n*” denotes the number of patients. In case of categorical variables, percentages are expressed between brackets. In case of continuous variables, mean value is reported. In case of nonnormal continuous variables, median value is reported, with interquartile range between brackets.

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
