# Peer review of "Impact of BRCA Mutation Status on Tumor Infiltrating Lymphocytes (TILs), Response to Treatment, and Prognosis in Breast Cancer Patients Treated with Neoadjuvant Chemotherapy"

_cancers, 2020, doi:10.3390/cancers12123681_

Round 1
Reviewer 1 Report
This is an interesting research, aiming at identify the correlations between BRCA mutation status and TILs and response to treatments. However, the authors divided the BRCA-deficient patients (n=46) further into breast cancer subtypes, having luminal (n=15), TNBC (n=27) and Her2+ (n=4). I don't think it is reliable to draw the conclusion that "pCR rates were significantly higher in BRCA-deficient BC and this association remained statistically significant only in the luminal BC subtype" based on the very small sample size. The authors are suggested to either include more patients in the study or be more cautious on making their conclusion due to the small sample size.
Author Response
We thanks reviewer 1 for this advices he/she suggested. We carefully reconsidered the whole manuscript and we applied the changes asked, notably while nuancing our conclusions regarding the small sample size of the population of the study.

Reviewer 2 Report
Beatriz Grandal 
et al. provided a retrospective analysis regarding the impact of BRCA mutation status on tumor infiltrating 
lymphocytes (TILs), response to treatment and 
prognosis in a cohort of breast cancer (BC) patients, over a decade of observations (2002-2012). In particular, the authors 
evaluated the role of BRCA across different BC subtypes in term of prognosis and prediction of response, in combination with the immune infiltration pre and post neo-adjuvant chemotherapy (NAC).
The major strength of the study lies on the fact this represents one of the largest cohort of BRCA-mut BC evaluation in the neo-adjuvant setting, combined with a dynamically TILs analysis pre/post NAC. The major limit of the study lies on the very nature of the retrospective, observational analysis, which eventually precludes any more accurate conclusion on the impact of BRCA in the neoadjuvant setting of early BC.
COMMENTS
- Introduction (line 96 and subseq): “…Hence, BRCA deficiency has sometimes 
been associated with a higher sensitivity to platinum agents when compared to other types neoadjuvant chemotherapy regimens [17–19]”. 
In a recent meta-analysis of platinum-based neoadjuvant chemotherapy in TNBC, among the BRCA-mutated patients included in RCTs, the addition of platinum was not associated with significantly increased pCR rate (OR 1.17, 95% CI 0.51–2.67, P = 0.711) (see Poggio F. et al, Ann of Oncol 2018). This observation has to be incorporated in the background of the manuscript
- Response to treatment (line 168 and subeq): “…BC subtype, this was confirmed only in the luminal BC subtype (33.3% (5/15), p= 0.006), but not in 
TNBC and HER2-positive BCs (48.1% (13/27), p= 0.823 and 75% (3/4), p= 0.291, respectively, Table S2, 
Figure 2). 
It might be useful to know the distribution of BRCA1 vs BRCA2 mutations in relation to the treatment’s response, across the different BC subtypes. Can the authors provide data on this? Based on the outcome of luminal BC, the authors speculate about the predictive role of BRCA-mut in HR+/HER2- as a possible criterion to support the use of NAC. However, the very nature of the study and the discrepancy b/w univariate and multivariate analyses should suggest caution in deriving conclusion on the role of BRCA in luminal BC as predictor of pCR after NAC.
- Post NAC Immune infiltration by BRCA status (line 188 and subseq and in othe part across the MS) Please always provide in the manuscript the crude numbers and not only percentages, to better appreciate the clinical significance of data
- Survival analysis (line 215 and subseq) RFS and OS were not significantly different b/w carriers of a 
BRCA pathogenic variant and BRCA-proficient patients, despite the different pCR rate observed (45.7% vs 28%). Please comment on this.
- Table 2. (line 234 and subseq) I suggest the retrospective studies (11) should be differentiated from prospective studies (6) in the layout of the table
- Discussion (line 291 and subseq) the description of the limits of the study should better highlight the possible biases related to the retrospective nature of the present analysis, especially for the implication on the patients’ survival outcome and response rate. A proper comment on this would be more than appreciated.
- Conclusions (line 362 and subseq) the authors generate the hypothesis the luminal 
BC patients with germline BRCA pathogenic variants may represent a subset of luminal BCs that are 
more likely to benefit from chemotherapy as primary treatment than the whole luminal BC 
population. 
However, the limited sample-size herein reported prevents any firm conclusions. We have to recognize the opportunity to select HR+/HER2- eBC for effective (neo)adjuvant chemotherapy still represents a matter of debate and one of the major advances in the field based on the use of MGAs. The authors are encouraged to consider this issue, in the context of the available data with a comment on this. (see Gianni L. et al J Clin Oncol 2005)
- Conclusions (line 369 and subseq) the authors hypothesize that since post-NAC immune infiltration seems to 
be higher in specimens of luminal tumors with BRCA pathogenic mutations, then those tumors would be more likely to respond to checkpoint inhibitors (IC) after 
chemotherapy. Do the authors can provide data on PD-L1 characterization to further support the statement? 
 This interesting hypothesis should be put in the context of other available data on the rationale for the use of IC in luminal BCs (see Anurag M et al. J Natl Cancer Inst 2020). Moreover, in BRCA-mut early BC it might be useful to mention data on the role of PARP-inhibitors in the (neo)adjuvant setting (see Goncalves at al Cancers 2020)
Author Response
We thank reviewer 2 for the time invested in reviewing and we really appreciated the accurate additionnal bibliography he provided. We think that his comments had a strong added value to our manuscript, and we included his suggestions in our revised report.

Round 2
Reviewer 2 Report
The authors properly reply to all of the comments made and provided a revised version of the manuscript embedding most suggestions in the main text, in the table, in the bibliography and in the supplementary materials.
The revised version of the manuscript significantly gained in clarity and scientific soundness and in my opinion could be consider for publication in its present form